# Report of Hermansky–Pudlak Syndrome in Two Families with Novel Variants in *HPS3* and *HPS4* Genes

**DOI:** 10.3390/genes14010145

**Published:** 2023-01-05

**Authors:** Qaiser Zaman, Muhammad Anas, Gauhar Rehman, Qadeem Khan, Aiman Iftikhar, Mashal Ahmad, Muhammad Owais, Ilyas Ahmad, Osama Yousef Muthaffar, Angham Abdulrhman Abdulkareem, Fehmida Bibi, Musharraf Jelani, Muhammad Imran Naseer

**Affiliations:** 1Department of Zoology, Government Postgraduate College Dargai, Malakand 23060, Khyber Pakhtunkhwa, Pakistan; 2Department of Zoology, Abdul Wali Khan University Mardan, Mardan 23200, Khyber Pakhtunkhwa, Pakistan; 3Mardan College of Medical Technologies, Mardan 23200, Khyber Pakhtunkhwa, Pakistan; 4Institute for Cardiogenetics, University of Luebeck, 23562 Luebeck, Germany; 5Department of Pediatrics, Faculty of Medicine, King Abdulaziz University, Jeddah 21589, Saudi Arabia; 6Faculty of Science, Department of Biochemistry, King Abdulaziz University, Jeddah 21589, Saudi Arabia; 7Center of Excellence in Genomic Medicine Research, King Abdulaziz University, Jeddah 21589, Saudi Arabia; 8Special Infectious Agents Unit, King Fahd Medical Research Centre, King Abdulaziz University, Jeddah 21589, Saudi Arabia; 9Department of Medical Laboratory Technology, Faculty of Applied Medical Sciences, King Abdulaziz University, Jeddah 21589, Saudi Arabia; 10Rare Diseases Genetics and Genomics, Centre for Omic Sciences, Islamia College Peshawar, Peshawar 25120, Khyber Pakhtunkhwa, Pakistan

**Keywords:** *HPS3*, *HPS4*, whole-exome sequencing, molecular diagnostics, Hermansky–Pudlak syndrome

## Abstract

*Background:* Hermansky–Pudlak syndrome (HSP) was first reported in 1959 as oculocutaneous albinism with bleeding abnormalities, and now consists of 11 distinct heterogenic genetic disorders that are caused by mutations in four protein complexes: AP-3, BLOC1, BLOC2, and BLOC3. Most of the patients show albinism and a bleeding diathesis; additional features may present depending on the nature of a defective protein complex. The subtypes 3 and 4 have been known for mutations in *HSP3* and *HSP4* genes, respectively. *Methods:* In this study, two Pakhtun consanguineous families, ALB-09 and ALB-10, were enrolled for clinical and molecular diagnoses. Whole-exome sequencing (WES) of the index patient in each family followed by Sanger sequencing of all available samples was performed using 3Billion. Inc South Korea rare disease diagnostics services. *Results:* The affected individuals of families ALB-09 and ALB-10 showed typical phenotypes of HPS such as oculocutaneous albinism, poor vision, nystagmus, nystagmus-induced involuntary head nodding, bleeding diathesis, and enterocolitis; however, immune system weakness was not recorded. WES analyses of one index patient revealed a novel nonsense variant (NM_032383.4: *HSP3*; c.2766T > G) in family ALB-09 and a five bp deletion (NM_001349900.2: HSP4; c.1180_1184delGTTCC) variant in family ALB-10. Sanger sequencing confirmed homozygous segregation of the disease alleles in all affected individuals of the respective family. *Conclusions:* The substitution c.2766T > G creates a premature protein termination at codon 922 in *HPS3*, replacing tyrosine amino acid with a stop codon (p.Tyr922Ter), while the deletion mutation c.1180_1184delGTTCC leads to a reading frameshift and a premature termination codon adding 23 abnormal amino acids to HSP4 protein (p:Val394Pro395fsTer23). To the best of our knowledge, the two novel variants identified in *HPS3* and *HPS4* genes causing Hermansky–Pudlak syndrome are the first report from the Pakhtun Pakistani population. Our work expands the pathogenic spectrum of *HPS3* and *HPS4* genes, provides successful molecular diagnostics, and helps the families in genetic counselling and reducing the disease burden in their future generations.

## 1. Introduction

Hermansky–Pudlak syndrome (HPS) was first described by two clinicians, Frantisek Hermansky and Paulus Pudlak, in 1959 and now has been classified into eleven subtypes based on different mutated genes and different clinical symptoms [1,2,3].

Beside the general symptoms of OCA and bleeding diathesis, the additional clinical manifestations of HPS vary with founder genes and their related protein complexes. For example, AP-3 defects lead to immunological anomalies, AP-3 and BLOC-3 deficiency leads to pulmonary fibrosis, and BLOC-3 deficiency causes gastrointestinal anomalies with variable degrees of severity [2].

The detailed clinical symptoms of HSP include oculocutaneous albinism such as hypo-pigmented white-to-light hair and skin, photosensitivity [4], nystagmus, decreased visual acuity, pale fundus, fovea hypoplasia, and iris transillumination [5,6]. In hematological anomalies, bleeding diathesis, easy bruising, epistaxis, menorrhagia, gingival bleeding, and colonic bleeding after trauma or surgery or postpartum are reported [2]. Respiratory problems such as pulmonary fibrosis, non-productive cough, exertional dyspnea, diffuse rales, and hypoxia are reported too [2,7,8,9]. In gastrointestinal anomalies, enterocolitis, abdominal pain, cramps, fever, weight loss, malabsorption, and frequent watery and bloody diarrhea are reported in the literature [8,10,11,12,13]. Neutropenia, immuno-deficiency, and frequent and recurrent infection are also reported in HPS [14,15,16,17].

HPS is caused by mutations in a set of genes which synthesize multi-subunit complexes such as AP-3, BLOC-1, BLOC-2, and BLOC-3. Each complex further consists of subunits having a role in different steps of membrane trafficking, vesicle transport, organelle maturations, and lysosomal-related organelle biogenesis pathways. Failure in the normal biology, due to variations in these genes’ products, could lead to a specific subtype of HPS. Mutations in the AP-3 complex cause HPS2 (AP3B1) and HPS10 (AP3D1), while mutations in BLOC-1 cause HPS7 (BLOC1S8), HPS8 (BLOC1S3), HPS9 (BLOC1S6), and HPS11 (BLOC1S5). The BLOC-2 complex is reported to cause HPS3 (BLOC2S1), HPS5 (BLOC2S2), and HPS6 (BLOC2S3). The mutation in the BLOC-3 complex causes HPS1 (BLOC3S1) and HPS4 (BLOC3S2) [2,3,15,17].

In the present study, two Pakhtun families were investigated for clinical and molecular diagnosis.

## 2. Material and Methods

### 2.1. Samples

The two enrolled families (ALB-09 and ALB-10) segregating autosomal recessive HPS belonged to the Pakhtun ethnic group of Khyber Pakhtunkhwa Province, Pakistan. Peripheral blood samples in EDTA tubes (5 mL) were collected from the available family members (Figure 1A,B).

### 2.2. Genomic DNA Extraction and Genetic Analysis

Genomic DNA from the available family members was extracted using QIAamp genomic DNA extraction kits (QIAGEN, Germantown, MD, USA). DNA was quantified on a Nanodrop 1000 spectrophotometer (Thermo Fisher Scientific, Waltham, MA, USA). Based on clinical manifestations and their causative genes involved in different kinds of HPS (Appendix A), the WES was performed for variant identification in index patients of the two families. Whole-exome sequencing covering approximately 22,000 human coding genes was performed via a commercial company: 3Billion Inc., Seoul South Korea (https://3billion.io/, accessed on 21 November 2021). Variant prioritization and candidate gene filtration were performed via data analyses pipelines of 3Billion Inc. along with EVIDENCE software as described earlier [18,19]. The shortlisted or potential candidate variants were subjected to Sanger sequencing in all family members. The target variants were PCR-amplified with forward and reverse primers (Table 1; Appendix A) and the purified products were sequenced using the CES facility of Macrogen Inc. South Korea (https://dna.macrogen.com/, accessed on 15 July 2022).

## 3. Results

### 3.1. Clinical Features of the Patients

#### 3.1.1. ALB-09 HPS3

In family ALB-09 presented here (Figure 1A), one affected brother (V-1) and two affected sisters (V-3 and V-5) were born to first-cousin marriage in the fifth generation, with yellowish-to-silver hair, fair white skin, light hazel eyes, nystagmus, head nodding, photophobia, poor vision acuity, night blindness, poor vision, skin photosensitivity, bleeding diathesis, and night gingival bleeding (Figure 2A,B). The parents and other family members could easily identify the affected child at the time of birth. All the affected showed slight changes in phenotypes with their ages; their hair became light brownish and then darkened, while small-sized freckles appeared on exposed areas of skin such as the shoulders, neck, face, and arms. No complaint of easy bruising or blood clotting was recorded in this family.

#### 3.1.2. ALB-10 HPS4

The second family, ALB-10, presented here (Figure 1B) is a large family with five affected individuals. Patients were born to first-cousin marriages in the fifth generation. All the affected (V-1, V-2, V-4, V-5, and V-6) showed a complete phenotype of HPS (Figure 2C,D). At birth, they had light-silver color hair; fair, pinkish-to-white skin; hazel eyes; nystagmus; severe head nodding; photophobia; light sensitivity; poor vision and acuity; night blindness; bleeding diathesis; and gastrointestinal abnormalities (enterocolitis). They could not hold their head and eyes in a static position, which worsened with age. In childhood, they were more albino as compared to their present adult ages. At the time of study, their ages were 6–23 years, they had golden hair, red skin, severe head nodding, photophobia, and rapidly expanding freckles on the exposed areas of skin such as the hands, face, neck, and shoulders. They had enterocolitis, gastrointestinal problems, abdominal pain, constipation, and cramp; otherwise, they were healthier and had no sign of bloody diarrhea. The female cousins (V-1 and V-2) were 14 and 16 years and had severe oculocutaneous albinism as compared to their male cousins (V-5 and V-6). The medical records of an affected individual (V-4) showed that he had died at 13 years due to severe bleeding, non-clotting of blood, and infection caused by tissue sampling for biopsy.

### 3.2. Genetic Analysis

WES identified two novel homozygous variants (NM_032383.4: c.2766T > G: p.Tyr922Ter) in the *HPS3* gene and (NM_001349900.2: c.1180_1184delGTTCC: p: Val394Pro395fsTer23) in the *HPS4* gene in families ALB-09 and ALB-10, respectively. In each family, Sanger validation confirmed autosomal recessive segregation of the identified variants with the disease phenotypes (Figure 3E,G). The variants were classified as “likely-pathogenic” according to the ACMG/AMP guidelines.

Both the variants were absent in homozygous form, in the ClinVar (https://www.ncbi.nlm.nih.gov/clinvar/, accessed on 12 October 2021), gnomAD v3.1.2 (https://gnomad.broadinstitute.org/, accessed on 12 October 2021), and 1000 genomes (https://www.internationalgenome.org/, accessed on 12 October 2021) databases. Population screening of 200 ethnically matched controls excluded the presence of these variants in healthy individuals.

## 4. Discussion

All those affected from both the families showed typical phenotypes of HPS including oculocutaneous albinism, as summarized in (Table 2).

A sum of 715 cases (Appendix A), with a prevalence rate of 1–9 per million, have been reported to date, and out of those, 333 cases have been reported from Puerto Rico and the Swiss Alps, making it the most affected ethnic group [2,20]. All over the world, a total of eleven types of HPS have been reported that are caused by mutations in AP-3, BLOC-1, BLOC-2, and BLOC-3 protein complexes, while from Pakistan, only three reports have been made so far that reported HPS1, HPS3, HPS4, HPS6, HPS8, and HPS9 types. The first report was a clinical study of two kindreds consisting of four affected individuals by Dr C Harrison in 2002 [20], while the second was a genetic study of an HPS-8 Pakistani-origin family with six affected individuals reported to have been caused by a novel biallelic mutation in *BLOC1S3* (c.448delC) [21]. A dedicated study of seven consanguineous Pakistani families by using WES analysis was conducted by [22], in which they reported seven pathogenic variants in five genes: *HPS1* (c.1342T > C, genomic deletion, c.2056C > T), *HPS3* (c.1509G > A), *HPS4* (c.276 + 5G > A), *HPS6* (c.823C > T), and *PLDN/HPS9* (c.232C > T).

The *HPS3* (OMIM: 606118) gene that is located on chromosome 3q24 contains 17 exons and encodes for a 1004 amino-acid-long protein, BLOC2S1, that interacts with HPS5 and HPS6 and is involved in the early stages of melanosome biogenesis and maturation [23,24]. A sum of 44 mutations have been reported in the *HPS3* gene: 16 missense, 10 splice substitution, 10 small insertion–deletion, 3 small deletion/duplications, 2 deletions, 2 gross deletion, and 1 gross insertion mutation; however, only 1 case of *HPS3* has been reported from Pakistan [22]. In our study of ALB-09, a novel homozygous substitution variant c.2766T > G (Figure 3A), in the *HPS3* (NM_032383.4) transcript at codon 922, has been identified that leads to the premature termination of HPS3 protein (NP_115759.2 p.Tyr922Ter) (Figure 3D). The substitution creates a nonsense variant, which is expected to cause a loss of normal protein function via nonsense-mediated mRNA decay and a loss of endoplasmic reticulum membrane retention signal KKPL (aa 1000–1003) motifs, as shown in Figure 4. The mutation c.2766T > G reported in *HPS3* is part of the BLOC-2 complex. The BLOC-2 complex has a significant role in the biogenesis and maturation of the late endosome and stage-II melanosome [2]. So, it is predicted that the mutation leads to non-biogenesis or hypo-biogenesis and the non-maturation of stage-II melanosomes and late endosomes (Figure 5). The effect is further aggravated in the affected individuals of ALB-9 by oculocutaneous albinism and bleeding diathesis.

The *HPS4* (OMIM: 606682) gene is located on chromosome 22q12.1 [25]. It has 14 exons encoding for 708 amino acids in the BLOC3S2 (OMIM: 606682) protein [26]. The BLOC-3 complex has a role in guanine exchange, the conversion of the inactive GDP-bound form to the active GTP-bound form, melanin production, melanosome biogenesis, and the promotion of membrane localization of RAB32 and RAB38. In *HPS4*, a total of 39 mutations have been reported that consist of 23 missenses, 5 splice substitutions, 5 small deletions, 5 small insertions/duplications, and 1 gross insertion. Just a single case of HPS4 has been reported from Pakistan [22]. In family ALB-10, a homozygous deletion c.1180_1184delGTTCC (Figure 3E, G) in the *HPS4* (NM_001349900.2) gene leads to a frameshift variant of two codons 394 and 395 (NP_001336829.1: p: Val394Pro395fsTer23) (Figure 3H) and the inducing of a stop codon 23 codons downstream from a point of deletion. The deletion mutation c.1180_1184delGTTCC in HPS4, part of the BLOC-3 protein complex, has an important role in Rab32/38-GDP conversion to Rab32/38-GTP. The activated Rab32/38-GTP has a role in stage-III melanosome maturation to stage-IV melanosomes and the maturation of late endosomes to platelet-dense granules, lamellar bodies, and lysosomes. The deletion creates a frameshift variant, which is expected to cause a loss of normal protein function via nonsense-mediated mRNA decay. The biallelic-affected individuals lack the normal HPS4 proteins, and their absenteeism results in the non-maturation of melanosomes (oculocutaneous albinism) and platelet-dense granules (bleeding diathesis) in family ALB-10.

## 5. Conclusions

The signs and symptoms of HPS have overlapping features with non-syndromic albinism and within HPS subtypes; in such cases, their exact clinical diagnosis could be challenging. WES has become the first-line molecular diagnostic test in rare Mendelian disorders. A population such as Pakistan where more than 60% of the population practice cousin marriages are at a high risk of autosomal recessive disorders. Premarital testing could be one of the best options to identify heterozygous carriers and educate them to avoid marriages in that case to reduce the disease burden in their future generations. This study broadens the pathogenic spectrum of *HPS3* and *HPS4* genes in these phenotypes. Proper genetic counselling, the introduction of a newborn-screening program, and parenteral diagnosis can play a major role in reducing the burden of such severe disorders [27]. This can be accomplished by performing newborn screening for all the common disorders and prenatal genetic testing for monogenetic disorders (PGT-M). PGT and in vitro fertilization are options for parents wishing to have future pregnancies [28,29,30].

## Figures and Tables

**Figure 1 genes-14-00145-f001:**
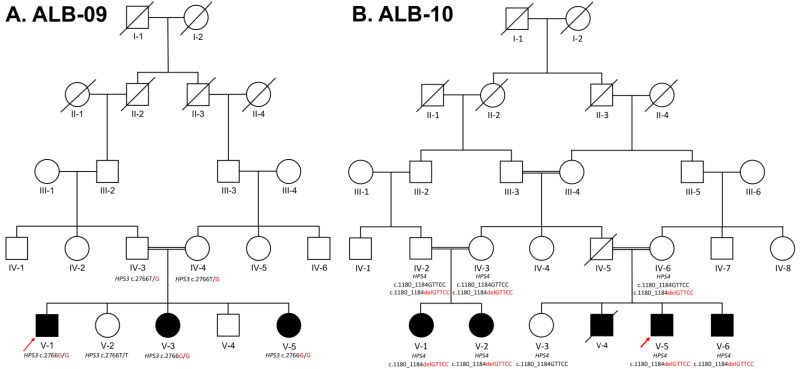
(**A**). ALB-09, five-generation pedigrees, showing autosomal recessive Hermansky–Pudlak syndrome type 3. The index patient V-1 is indicated with an arrow, and the genotypes of those family members are written below their symbols whose samples were available for Sanger sequencing. The three affected siblings (V-1, V-3, and V-5) have homozygous alleles, while one unaffected sibling (V-2) has homozygous wild-type alleles, and their parents (IV-3 and IV-4) have heterozygous alleles for HPS3 c.2766T > G. (**B**). ALB-10, a five-generation pedigree, presenting autosomal recessive Hermansky–Pudlak syndrome type 4 in two different clades while the index patient V-5 is indicated with an arrow. Both clades were confirmed for the variant in the HPS4 gene, and Sanger sequencing was carried out and confirmed that those affected (V-1, V-2, V-5, and V-6) were homozygous, the parents (IV-2, IV-3, and IV-6) were heterozygous, and the normal sibling (V-3) was homozygous wild-type for c.1180_1184delGTTCC, which are written below each processed member of the family.

**Figure 2 genes-14-00145-f002:**
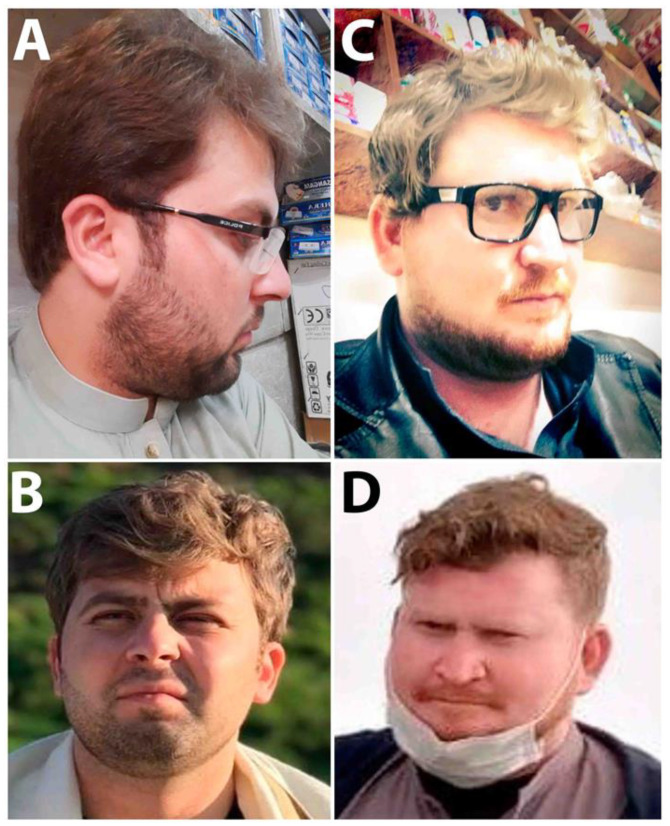
Clinical observations of ALB-09 (**A**,**B**) and ALB-10 (**C**,**D**) show hypopigmented eyes, skin, hair, and photophobia while no sign of easy bruising on the skin was observed.

**Figure 3 genes-14-00145-f003:**
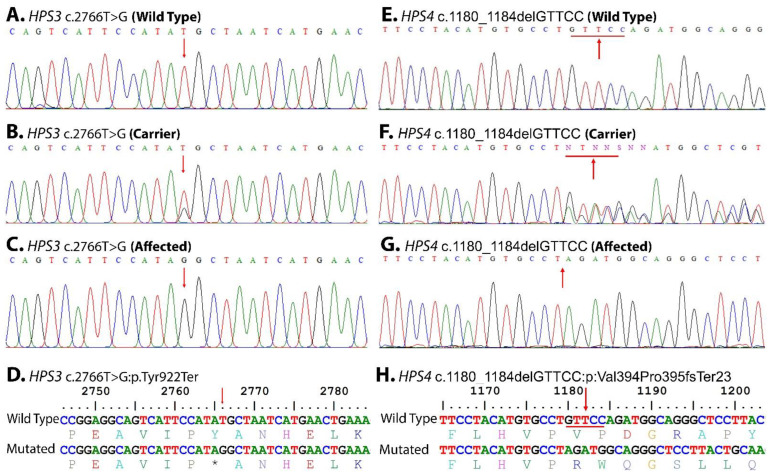
Sanger sequence trace files identified homozygous mutated, heterozygous carriers, hemizygous affected, and homozygous wild type for each gene, and the point of alteration is indicated by an arrow above the trace files as (**A**–**D**): HPS3 c.2766T > G, (**E**–**H**): HPS4 c.1180_1184delGTTCC. One litter translation of wild type and mutant confirms the altered proteins.

**Figure 4 genes-14-00145-f004:**
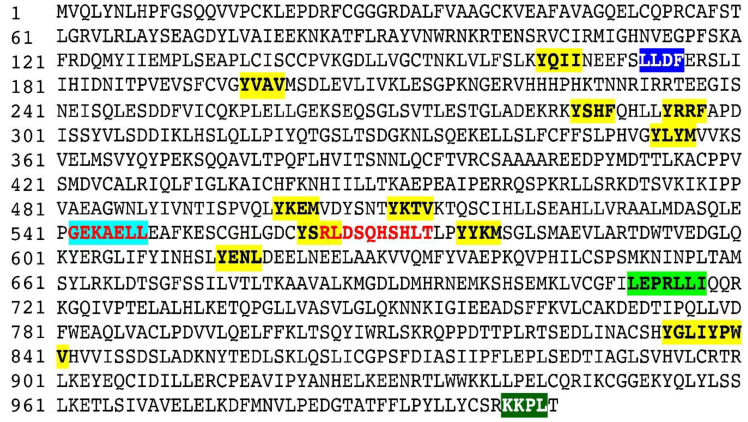
HPS3 protein contains clathrin-binding motif LLDF (aa 172–174), endoplasmic reticulum membrane retention signal KKPL (at 1000–1003), 2 dileucine-based sorting motifs GRKAELL (aa 542–548) and LEPRALLI (aa 711–717), 12 Tyrosine Sorting motifs, and 1 peroxisomal matrix targeting signal RLDSQHSHLT (564–573).

**Figure 5 genes-14-00145-f005:**
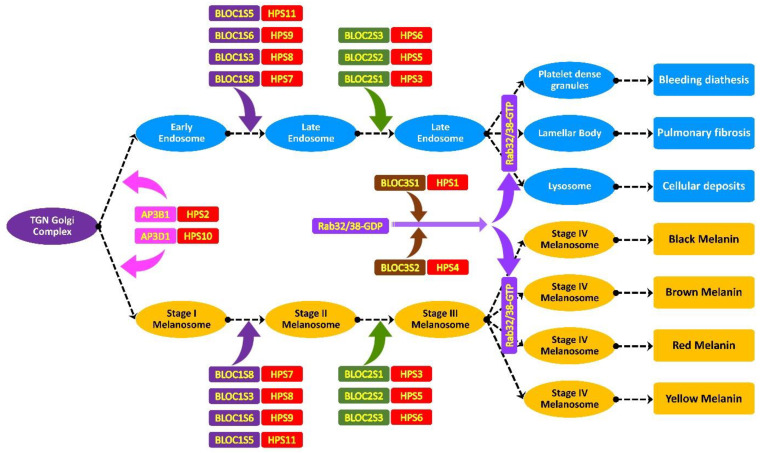
Endosomal pathway of TGN (trans-Golgi network) showing different stages of melanosome maturation, platelet δ-granule (dense granule), lamellar body and lysosome formation are shown. Protein complexes AP-3 (adaptor protein), BLOC-1 (biogenesis of lysosome-related organelles complex-1), BLOC-2 (biogenesis of lysosome-related organelles complex-2) and BLOC-3 (biogenesis of lysosome-related organelles complex-3) are shown. Any mutation in these complexes or in their subunits result in a different class of Hermansky–Pudlak syndrome.

**Table 1 genes-14-00145-t001:** Primers designed by Primer3plus, written with gene name, exon no, annealing temperature, and product size.

S/No	Gene	Exon	Primers (5′-3′)	Temp	Product Size (bp)
1	*HPS3*	Exon 15	F 5′-TTGGCAAGTGAGGTTTAGTCTT-3′	56 °C	383 bp
			R 5′-TTGCAGGAAAGGAAATGAGG-3′		
2	*HPS4*	Exon 11	F 5′-GCAGGACTGCACAACTCTG-3′	57 °C	637 bp
			R 5′-CCTTTGACGCAGTGAGTGTA-3′		

For primers’ selection, the genomic DNA sequence of *HPS3* was retrieved through Ensembl accession number ENSG00000163755 with transcript ID ENST00000296051.7 and *HPS4* via ENSG00000100099 with transcript ID ENST00000398145.7.

**Table 2 genes-14-00145-t002:** Phenotypic comparison of HPS3 and HPS4 patients with the previously reported patients.

S/No	Clinicals/Phenotypes with Reported References	HPS3	HPS4	ALB-09	ALB-10
1	Ocular albinism	+	+	+	+
2	Reduced visual acuity	+	+	+	+
3	Horizontal nystagmus	+	+	+	+
4	Photophobia	+	+	+	+
5	Iris transillumination	+	+	+	+
6	Hypopigmentation of retina	+	+	+	+
7	Foveal hypoplasia	+	+	n/a	n/a
8	Skin pigment dilution, mild-to-severe, relative to unaffected family members	+	+	+	+
9	Weakly pigmented basal cell layer	n/a	+	n/a	n/a
10	Normal number of melanocytes	-	+	n/a	n/a
11	Reduced amount of melanin pigment in melanocytes	-	+	+	+
12	Accumulation of ceroid pigment in perivascular macrophages	-	+	+	+
13	Hair pigment, mild-to-severe, dilution relative to unaffected family members	+	+	+	+
14	Bleeding tendency	+	+	+	+
15	Easy bruising	+	+	-	-
16	Gingival bleeding	+	+	+	+
17	Epistaxis (nose bleeding)	-	+	+	+
18	Menorrhagia	+	+	+	+
19	Absence of platelet-dense bodies	+	+	n/a	n/a
20	Lack of secondary aggregation response of platelets	+	+	n/a	n/a
21	Pulmonary fibrosis	-	+	-	+
22	Restrictive lung disease	-	+	n/a	n/a
23	Enterocolitis	-	+	+	+
24	Reported gene	HPS3/BLOC2S1	HPS4/BLOC3S2	HPS3/BLOC2S1	HPS4/BLOC3S2

## Data Availability

The datasets used and/or analyzed during the current study are available from the corresponding author upon reasonable request for research only.

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
