# Peer review of "Report of Hermansky–Pudlak Syndrome in Two Families with Novel Variants in HPS3 and HPS4 Genes"

_genes, 2023, doi:10.3390/genes14010145_

Round 1
Reviewer 1 Report
In this study, authors identified Hermansky Pudlak Syndrome in two families with novel variants in HPS3 and HPS4 genes. Whole exome sequencing (WES) of the index patient in each family, followed by Sanger sequencing was done. WES analyses of one index patient revealed a novel nonsense variant (NM_032383.4: HSP3; c.2766T>G) in family ALB-09 and a five bp deletion (NM_001349900.2: HSP4; c.1180_1184del variant in family ALB-10. Sanger sequencing confirmed homozygous segregation of the disease alleles in all affected individuals of the respective families. This is well written and interesting study, but I have a few comments.
Comments:
1-Please italicize gene names throughout the manuscript.
2-WES protocol has been described very briefly. Kindly elaborate on the procedure of WES.
3- Mention transcript ID with mutation identified.
4- Did the authors find any genotype-phenotype correlations?
5- Cite the new release of HGMD i.e. 2019.4, and mention its updated information
6- In the methodology section, please provide the primer sequence used for the PCR and Sequencing validation.
7- The analysis strategy is not in sequence, it should be in this way that fast files were converted to Bam, then BAM to vcf, and then variant filtration was carried out on this file, etc needed in detail.
8- Figure legends are not written properly, and what the figure represents is also unclear.
9- Please add the latest reference related to the study and correct the format of the reference as some of the references are missing also.
10- Details of ethical approval are missing.
11- Improve the quality of the figures.
12-Language also needs corrections as some of the sentences in the introduction and discussion are unclear.
The manuscript is well-written, however, it still needs improvement before the final acceptance for publication.
Author Response
Comments of Reviewer 1
In this study author’s identified Hermansky Pudlak Syndrome in two families with novel variants in HPS3 and HPS4 genes. Whole exome sequencing (WES) of the index patient in each family followed by Sanger sequencing was done. WES analyses of one index patient revealed a novel nonsense variant (NM_032383.4: HSP3; c.2766T>G) in family ALB-09 and a five bp deletion (NM_001349900.2: HSP4; c.1180_1184del variant in family ALB-10. Sanger sequencing confirmed homozygous segregation of the disease alleles in all affected individuals of the respective families. This is well written and interesting study but I have few comments before it will get accepted for publication.
We are thankful for the reviewer for the review our manuscript and suggestions to improve the overall quality of the manuscript. Thanks
Comments:
- Please italicize gene names throughout the manuscript.
Response to the comments: We have corrected the manuscript as suggested by the reviewer.
2-WES protocol has described very briefly. Kindly elaborate the procedure of WES.
3- Mention transcript ID with mutation identified.
Response to the comments: The Transcript ID has been mentioned in the manuscript as suggested.
4- Did authors find any genotype-phenotype correlations?
Response to the comments: We did not find any genetotype-phenotype difference. Our identified samples showed no difference of disease phenotype as previously identified cases related to the disease.
5- Cite the new release of HGMD i.e. 2019.4 and mention its updated information’s
Response to the comments:
6- In the methodology section please provide the primer sequence used for the PCR and Sequencing validation.
Response to the comments: Table 1 in the manuscript showing the primer sequence used for this study.
7- Analysis strategy is not in sequence, like, it should be in this way that fastq files were converted to Bam, then BAM to vcf and then variant filtration was carried out on this file etc need in detail.
Response to the comments: We already followed the standard guidelines for data analysis. Thanks
8- Figure legends is not written properly and what figure is representing what is also not clear.
Response to the comments: We have revised the figure legend by adding more details to make it clear. Thanks
9- Please add the latest reference related to the study and correct the format of reference as some of reference are missing also.
Response to the comments: We have added the latest available reference related to the study. Thanks
10- Details of ethical approval is missing.
Response to the comments: ethical approval number has been added in the revised manuscript. Informed consent was provided by the family elders before the start of this study and was ap-proved by Institutional Bioethics Committee (IBC), Islamia College Peshawar (ref. # 529/oric/icp).
11- Improve the quality of the figures.
Response to the comments: We have improved the quality of figures. Thanks
12-Language also need corrections as some of the sentences in the introduction and discussion is not clear.
Response to the comments: The language of the manuscript has improved with help of native English speaker.
Over all the manuscript is well writing need improvement before the final acceptance for publication.
Thanks for the reviewer’s positive comments to improve the quality of the manuscript.

Reviewer 2 Report
Dear Editor,
Thank you for inviting me as a reviewer. Following the review of the review article, entitled, “The first report of Hermansky Pudlak Syndrome in two families with novel variants in HPS3 and HPS4 genes”, by Zaman et al., I recommend that it should be revised taking into given below suggestions.
1. The data presented in this study can be strengthened with supporting protein modeling evidence. The author should add the effect of these mutations on the structure, stability, and function of related proteins after conducting protein modeling.
2. Add information about the effect of these mutations on the lysosome-related organelles (LROs).
Abstract:
1. Use the HGSV nomenclature for missense variant.
Introduction:
1. Add a few lines explaining the genetic architecture of Hermansky Pudlak Syndrome, as in its current form, most of the information mentioned is related to medical manifestation.
Methods:
1. Add some details on the diagnosis of the study subjects. How the diagnosis was conducted? Was it phenotypic only?
2. The text in lines 106-108 is confusing, please elaborate.
3. Mention the genomic build number used to design primers.
4. Mention the date when the sequence used for designing primers was accessed.
5. Mention the “Gene ID” and “Transcript ID” used to design primers.
6. I would strongly suggest providing the sequence of exon 15 (HPS3), and exon 11 (HPS4) in the supplementary and highlighting the primer binding sites.
7. Did you use bidirectional sanger sequencing?
Results:
1. Figure 3F: It seems like there are a few additional mutations in the carrier subject. Did you explore that?
2. Revise the title of Table 2.
Discussion:
1. As the author mentioned that earlier reported gene mutation associated with Hermansky Pudlak Syndrome. Did the author observe any of those earlier reported mutations in their current data set? If yes, please mention the comparison of the current study with earlier reported mutations.
Sincerely,
Reviewer
Author Response
Comments of Reviewer 2
Thank you for inviting me as a reviewer. Following the review of the review article, entitled, “The first report of Hermansky Pudlak Syndrome in two families with novel variants in HPS3 and HPS4 genes”, by Zaman et al., I recommend that it should be revised taking into given below suggestions.
- The data presented in this study can be strengthened with supporting protein modeling evidence. The author should add the effect of these mutations on the structure, stability, and function of related proteins after conducting protein modeling.
Response to the comments: We highly appreciate the suggestion of the review expert. But, the reason that why Protein Modeling has not been performed, is that the two novel homozygous variants (NM_032383.4: c.2766T>G: p.Tyr922Ter) in HPS3 gene and (NM_001349900.2: c.1180_1184delGTTCC: p: Val394Pro395fsTer23) in HPS4 gene in families ALB-09 and ALB-10 respectively, are premature terminations. Whenever two homozygous premature termination codons come they are not able to produce a stable protein due to non-mediated mRNA decay process. That is why protein modeling could not be performed for these two cases.
- Add information about the effect of these mutations on the lysosome-related organelles (LROs).
Response to the comments: A text from Lines # 215-221 has been added “The mutation c.2766T>G reported in HPS3, is the part of BLOC-2 complex. The BLOC-2 complex has significant role in biogenesis and maturation of late endosome and stage-II melanosome (Yousaf et al., 2016; Huizing et al., 2020). So, it is predicted that the mutation leads to non-biogenesis or hypo biogenesis and non-maturation of stage-II melanosomes and late endosomes (Fig. 6). The effect is further advocated in the affected individuals of ALB-9 by oculocutaneous albinism and bleeding diathesis.”
Lines # 237-245, have been added and the confusion has been removed. “The deletion mutation c.1180_1184delGTTCC in HPS4, part of BLOC-3 protein complex, has important role in Rab32/38-GDP conversion to Rab32/38-GTP. The activated Rab32/38-GTP has role in stage-III melanosomes maturation to stage-IV melanosomes and maturation of late endosomes to platelets dense granules, lamellar bodies, and lysosomes (Fig. 6). The deletion creates a frameshift variant, which is expected to cause a loss of normal protein function via nonsense-mediated mRNA decay. The biallelic affected individuals lack the normal HPS4 proteins, and their absenteeism results in non-maturation of melanosomes (oculocutaneous albinism) and platelets dense granules (bleeding diathesis) in family ALB-10.”
Abstract:
- Use the HGSV nomenclature for missense variant.
Response to the comments: The variants have been corrected in abstract and at other places wherever required according to HGSV format.
Introduction:
- Add a few lines explaining the genetic architecture of Hermansky Pudlak Syndrome, as in its current form, most of the information mentioned is related to medical manifestation.
Response to the comments: A text in introduction has been added, “HPS are caused by mutation in a set of genes, which synthesize multi-subunit complexes like AP-3, BLOC-1, BLOC-2, and BLOC-3. Each complex further consists of subunits having role in different steps of membrane trafficking, vesicles transport, organelles maturations and lysosomal related organelles biogenesis pathways. Failure in the normal biology, due to variations in these genes’ products, could lead to a specific subtype of HPS. Mutations in AP-3 complex causes HPS2 (AP3B1) and HPS10 (AP3D1), while mutation in BLOC-1 causes HPS7 (BLOC1S8), HPS8 (BLOC1S3), HPS9 (BLOC1S6) and HPS11 (BLOC1S5). BLOC-2 complex is reported to cause HPS3 (BLOC2S1), HPS5 (BLOC2S2) and HPS6 (BLOC2S3). The mutation in BLOC-3 complex causes HPS1 (BLOC3S1) and HPS4 (BLOC3S2) (Ammann et al., 2016; Okamura et al., 2018; Huizing et al., 2020; Boeckelmann et al., 2021)).
Methods:
- Add some details on the diagnosis of the study subjects. How the diagnosis was conducted? Was it phenotypic only?
Response to the comments: Yes, initially we have selected the families based on phenotypes, and performed WES. Additional to the phenotypes, we have investigated their clinical history by their previous treatment and abnormalities.
- The text in lines 106-108 is confusing, please elaborate.
Response to the comments: These sentences have been revised to remove confusions.
- Mention the genomic build number used to design primers.
Response to the comments: The Ensembl Genome Browser (https://asia.ensembl.org/Homo_sapiens/Info/Index) build GRCh38.p13 had been used for designing/selecting primers. The website was accessed last time in July 2022.
- Mention the date when the sequence used for designing primers was accessed.
Response to the comments: The Ensembl Genome Browser (https://asia.ensembl.org/Homo_sapiens/Info/Index) build GRCh38.p13 had been used for designing/selecting primers. The website was accessed last time in July 2022.
- Mention the “Gene ID” and “Transcript ID” used to design primers.
Response to the comments: For primers selection the genomic DNA sequences of HPS3 were retrieved through Ensembl accession number ENSG00000163755 with transcript ID ENST00000296051.7 and HPS4 via ENSG00000100099 with transcript ID ENST00000398145.7. This text has been added below the primers table for its immediate reference.
- I would strongly suggest providing the sequence of exon 15 (HPS3), and exon 11 (HPS4) in the supplementary and highlighting the primer binding sites.
Response to the comments: A new world file by the name of SUPPLEMENTARY DATA 1 is attached, that has the genomic/CCDS sequence of the gene, altered point, reverse and forward primers separately-cum-highlighted and their product size.
- Did you use bidirectional sanger sequencing?
Response to the comments: Yes, bidirectional Sanger sequencing was performed.
Results:
- Figure 3F: It seems like there are a few additional mutations in the carrier subject. Did you explore that?
Response to the comments: Actually in carriers’ samples five bases deletion could lead to generate sequencing data reading of two alleles i.e. the one with wild type sequence and the other with mutant sequence. That is why it creates confusions that it may contain something else. However, we have thoroughly checked there is nothing extra than we have reported already.
- Revise the title of Table 2.
Response to the comments: The title has been revised, “Phenotypic comparison of HPS3 and HPS4 patient with the previously reported patients.”
Discussion:
- As the author mentioned that earlier reported gene mutation associated with Hermansky Pudlak Syndrome. Did the author observe any of those earlier reported mutations in their current data set? If yes, please mention the comparison of the current study with earlier reported mutations.
Response to the comments: None of the previous mutation has been found in our current data set in WES.

Reviewer 3 Report
In this paper the authors report two families with Hermansky-Pudlak syndrome (HPS) and describe novel mutations found in the background of the disease. Clinical description of the cases are well-written together with the results of the genetic analysis. However, lack of laboratory investigation of platelets is a major drawback of this case study.
There are some issues in addition to be solved, as follows,
The first part of the discussion is not necessary, since it repeats the pieces of information already given in the results section.
English should be improved, and typos should be eliminated, like in the title of Table 2. “Details” is written instead of “detailed”, in 9th raw of Table 2. “n/r” is written instead of “n/a”, in 7th page (raw 206) “in deletions” is written instead of “insertion-deletion”, etc.
Author Response
Comments of Reviewer 3
In this paper the authors report two families with Hermansky-Pudlak syndrome (HPS) and describe novel mutations found in the background of the disease. Clinical description of the cases are well-written together with the results of the genetic analysis. However, lack of laboratory investigation of platelets is a major drawback of this case study.
Response to the comments: We agree with the suggestion of the subject expert/reviewer; however, as the patients belonged to remote village of Khyber Pakhtunkhwa and we could not find the required tests in the nearby hospitals. Furthermore, as we have clearly mentioned in the clinical results of family ALB-9 that we have recorded “bleeding diathesis, night gingival bleeding”. In case of family ALB-10, bleeding diathesis was recorded. One member (V-4) had died at the age of 13 due to severe bleeding, non-clotting of blood and infection caused by tissue sampling for biopsy, is an indication of bleeding and clothing abnormalities.
There are some issues in addition to be solved, as follows,
The first part of the discussion is not necessary, since it repeats the pieces of information already given in the results section.
Response to the comments: The first paragraph of discussion has been removed as suggested by the reviewer.
English should be improved, and typos should be eliminated, like in the title of Table 2. “Details” is written instead of “detailed”, in 9th raw of Table 2. “n/r” is written instead of “n/a”, in 7th page (raw 206) “in deletions” is written instead of “insertion-deletion”, etc.
Response to the comments: English errors have been removed with great care.

Round 2
Reviewer 1 Report
Agreed with the improved mansucript
Author Response
We are thankful for the reviewers comment to improve the manuscript.
Reviewer 2 Report
The author has addressed all the points.
Author Response
The author has addressed all the points.
Response: We are thankful for the reviewers comments to improve the manuscript.
Reviewer 3 Report
The authors have submitted the revised version of the manuscript. I understand that the laboratory investigations are impossible to be performed and added to the manuscript.
However, there are still typos throughout the text including the abstract, where "HPS" is mixed up with "HSP" (HPS is the correct name of the genes, please, correct, where needed). First sentence of the new discussion part is also needs attention. Title of table 2 should also be corrected (ie "patients" should be written instead of "patient"). Line 280:"parental" should be written instead of "parenteral". Please go throughout the text once again, carefully.